# Possible Mechanism for the Tsunami-Related Fires That Occurred at Aonae Harbor on Okushiri Island in the 1993 Hokkaido Nansei-Oki Earthquake

**Yuji Enomoto [1],*, Tsuneaki Yamabe [1], Shigeki Sugiura [2] and Hitoshi Kondo [2]**

[1]   Fii Center, Shinshu University, Ueda Campus, 3-15-1 Tokida, Ueda, Nagano 386-8567, Japan;
      yamabe-t@shinshu-u.ac.jp
[2]   Genesis Research Institute, Inc., 4-1-35 Noritake-Shinmachi, Nishi-ku, Nagoya, Aichi 451-0051, Japan;
      s-sugiura@konpon.co.jp (S.S.); hkondo@konpon.co.jp (H.K.)
*    Correspondence: enomoto@shinshu-u.ac.jp

**Abstract:** In this paper, we investigate the mysterious tsunami fires that occurred at Aonae Harbor on Okushiri Island during the 1993 Hokkaido Nansei-Oki earthquake. Specifically, five fishing boats moored separately from each other in the harbor suddenly caught fire and burned nearly simultaneously with the arrival of the first tsunami wave. However, the ignition mechanism of those fires has, until now, remained largely unknown. At the time the earthquake occurred, an NHK (Japan Broadcasting Corporation, Tokyo, Japan) crew that was on the island to report on its scenic natural attractions just happened to capture video footage of those tsunami-related fires. Using that NHK video footage in combination with eyewitness accounts, this study investigates the spatio-temporal process leading to those tsunami-related fires. For example, one witness said, "There was whitish bubbling in the offshore area and I saw five burning fishing boats moored on the seawall being blown about by the strong winds. The burning boats were swept ashore with the tsunami and ignited the gasoline of a car that was rolling in the waves. The fire eventually spread to the center of the Aonae District." The NHK video footage confirmed flames arising from the five fishing boats almost simultaneously and the shimmering white color of the tsunami waters striking the seawall, which were consistent with the eyewitness testimony. Based on these spatio-temporal data, we propose the following hypothetical model for the origin of tsunami fires. Combustible methane gas released from the seabed by the earthquake rose toward the surface, where it became diffused into the seawater and took the form of whitish bubbles. The tsunami strike on the Aonae Harbor seawall resulted in the generation of large electrical potential differences within the seawater mist, which quickly developed sufficient electrical energy to ignite the methane electrostatically. The burning methane bubbles accumulated on the boat decks, which then burned violently.

**Keywords:** tsunami fire; 1993 Hokkaido Nansei-Oki earthquake; electrostatic ignition; methane bubbles; Okushiri Island; Aonae Harbor

---

## 1. Introduction

The M7.8 Hokkaido Nansei-Oki earthquake that occurred on 12 July, 1993 at 22:17 local time, produced a devastating tsunami wave that quickly hit Okushiri Island [1–6] (see Figure 1a–f). Approximately 4 min after the mainshock, a tsunami that was reported to be approximately 10 m high struck and swept away much of the urban area of the island's Aonae Cape. When the first tsunami wave was reflected in Aonae Bay due to the lens effect between the Aonae Cape south shallows and the Okushiri Spur [3] (see Figure 1b), people who had evacuated to the nearby hills of the Aonae District witnessed the five fishing boats moored at the harbor catching fire and burning

simultaneously (see Figure 2a). Eventually the boat fires were spread to the shore by the gusting winds that accompanied the second tsunami wave, where they ignited propane gas released from storage cylinders, gasoline from the cars, and kerosene from home-heating fuel tanks. The resulting conflagration completely destroyed much of the Aonae urban area (see Figure 3a). Later, a survey team of the Japan Society of Civil Engineers (JSCE) investigated the damage caused by the tsunami-related fires [7] but were unable to conclusively determine their origins.

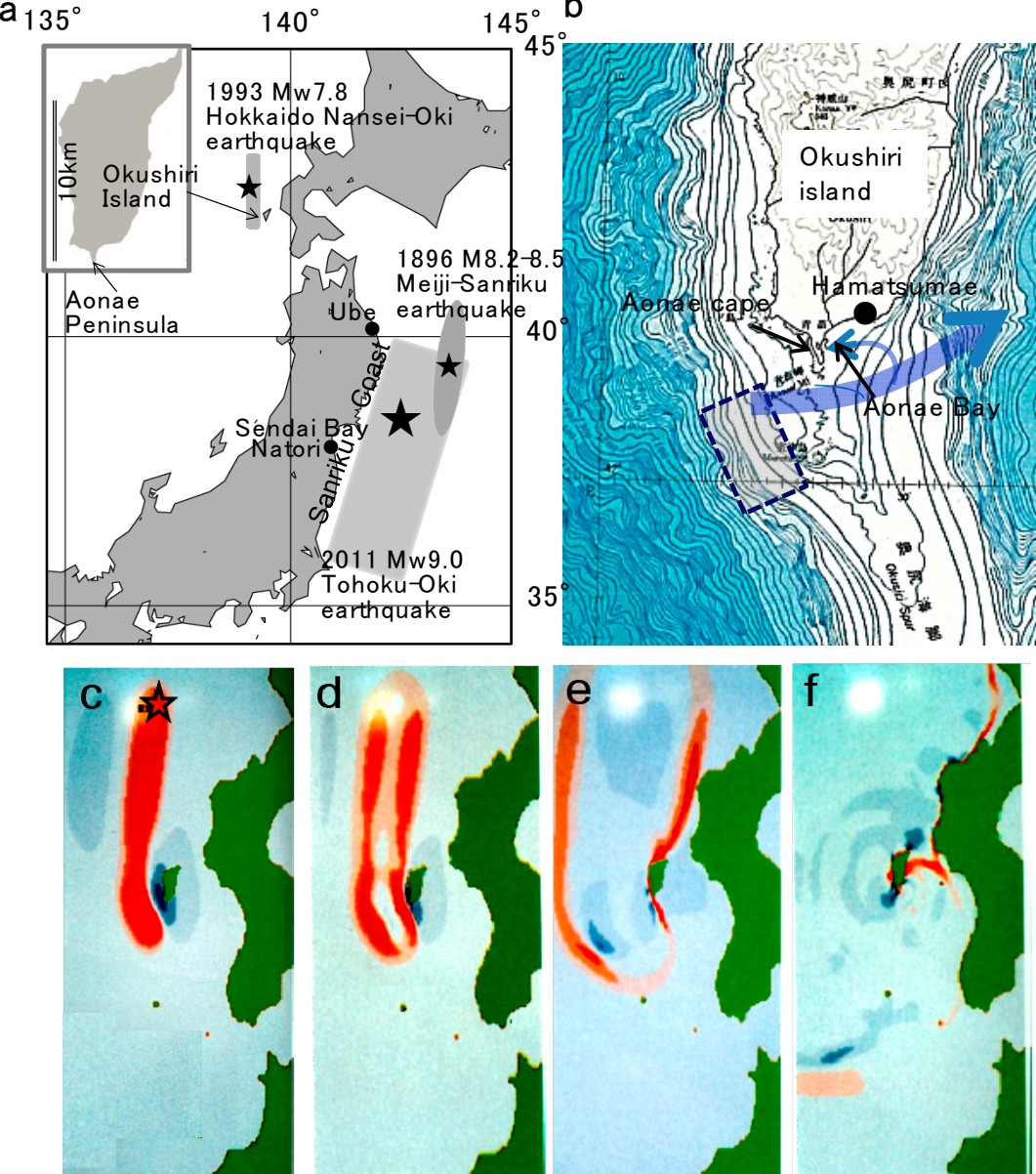

**Figure 1.** (**a**) Map showing the epicenter of the earthquakes where the tsunami-related fires occurred. Stars show the main shock epicenters. (**b**) Map of south area of Okushiri Island (created by the Japan Coast Guard). Arrows indicate the path of the first tsunami wave, a dotted square shows the survey area by the Japan Agency for Marine-Earth Science and Technology (JAMSTC). (**c–f**) Computer graphics of tsunami propagation: Immediately after the fault rupture, and 1, 4, and 12 min after the fault rupture, respectively [6], A star in (c) shows the epicenter. (courtesy of Tohoku University Disaster Control Center and Okushiri Town Hall).

The 2011 $M_w$9.0 Tohoku-Oki earthquake caused numerous tsunami-induced fires over a wide area of the Sanriku Coast (see Figure 1a). Since then, concern over tsunami-induced fires has grown markedly, and several studies have analyzed the occurrence tendency [8,9], assessed the damage caused and hypothesized on an initiation mechanism [10,11], and simulated such fires to improve preparedness [12]. According to a comprehensive survey by the Japan Association for Fire Science and Engineering [13], a total of 159 tsunami-related fires occurred in the aftermath of the 2011 Tohoku-Oki earthquake. Of these, the specific origins of those fires (such as vehicles, electricity meters, and power panels) could be determined only in 56 cases, which means that, for the other 103 cases, the fire origins remain unknown.

Scientific investigations into the possible mechanisms of tsunami-induced fires are essential from the standpoint of disaster prevention. Unfortunately, when investigating events that have occurred years in the past, it is often difficult to find sufficient clues to elucidate those mechanisms from eyewitness testimonies and media reports. Furthermore, although it is possible to understand, to some extent, the conditions that existed based on such testimony and reports, those accounts are normally insufficient for gaining a spatio-temporal understanding of tsunami-related fires and determining their causes.

However, in the case of the 1993 Hokkaido Nansei-Oki earthquake and the tsunami-induced fires that struck Okushiri Island, we recently found interesting clues in a NHK TV video taken during those fires. On the day of the earthquake, an NHK TV crew just happened to be on the island to conduct interviews and film the natural wildlife of Okushiri Island. A portion of those video recordings was aired on the "NHK Special" program four days after the earthquake. Unfortunately, since the video footage was not immediately made public afterward, it was not utilized as research material in the JSCE survey team or by other researchers at that time. The present research could obtain access to and use the raw video footage taken by the NHK crew.

In our present work, we engaged in organizing eyewitness information reported by the media as summarized elsewhere [14], the NHK TV video footage, and a computer simulation of the tsunami propagation [5,6] (see Figure 1c–f), in order to investigate the spatio-temporal process leading to the abovementioned tsunami-related fires. We then formed a working hypothesis in which we posited that electrostatic energy provided the initial sparks that ignited the fires and proposed a possible mechanism. We also report on laboratory experiments conducted to validate our hypothesis.

## 2. Materials and Method

### 2.1. Materials for Spatio-Temporal Events Study

In a normal earthquake investigation, it is first necessary to organize a spatio-temporal series of events that occurred after the mainshock. However, in the case of the 1993 Hokkaido Nansei-Oki earthquake, the time history of the tsunami-related events on Okushiri Island was not precisely known, because no tidal records exist. Accordingly, it was necessary to reconstruct the time history leading to the tsunami-related fires using somewhat unreliable witness testimony obtained via a series of interviews. To facilitate this, we collected various published accounts related to the nature of the tsunami and tsunami-related fires from newspapers, magazines, books, and other reports in order to gather as much data as possible. The collected testimonies, which were reported elsewhere, were all in Japanese [8].

One such source was the NHK TV program entitled "*Oo-tsunami ga osotta, Okushiri-tō karano Houkoku*" (in Japanese), which means "Report from Okushiri Island where the great tsunami hit," that was broadcast on July 16, 1993, from 20:00 to 20:50. Of this 50 min program, the video footage dealing with the tsunami-related fires was just 2 min and 50 s long. Video footage of the boat fire scenes, which was recorded from Point C in Figure 2a, began about 10 min after the earthquake.

The computer graphic (CG) images on the tsunami propagation created by Tohoku University Disaster Control Center (see Figure 1c–f) [5], which were included in the "*Hokkaido Nansei-Oki*

*Jisin Okushiri-cho Kirokusho*" (Okushiri-cho ed.) [6] were also helpful for understanding the tsunami propagation process.

## 2.2. Laboratory Experiments

Based on the spatio-temporal events leading to the boat fires, we developed a working hypothesis for the possible mechanism of the tsunami fire, in which methane-bearing bubbles carried by tsunami-induced winds struck the seawall, resulting in the electrostatic ignition of the gaseous methane, as will be described in detail later. In order to investigate the feasibility of this hypothesis, we measured the electric current/potential of splashed droplets generated by a falling weight on seawater suffused with gaseous methane. The details of this experimental setup are described in Appendix A.

## 3. Results and Discussion

### 3.1. Spatio-Temporal Survey of the Events Leading to the Tsunami Fires

Based on the NHK TV video and the eyewitness testimonies, the spatio-temporal events related to the tsunami-related fires are summarized as follows:

**July 12**

22:17—The Hokkaido Nansei-Oki earthquake (M7.8) occurred in the back-arc region of the convergent boundary where the Pacific Plate subducts beneath the Eurasian Plate (see Figure 1a). The epicenter was approximately 60 km off of Okushiri Island along the west coast of Hokkaido (see Figure 1a,c).

22:21—About 4 min after the mainshock, the first tsunami wave (approximately 10 m high) struck Aonae District 5, washing away much of that urban area even though it was protected by a 4.5 m high seawall. Regarding that moment, T. Hasegawa, who was trapped in a car at Point A in Figure 2a, said, "I saw the second floor of a house covered by the tsunami water and a whitish splashing wave shining above the roof." Since all of the island's electricity had stopped soon after the earthquake, his report indicates that the shimmering phenomenon was visible in the dark. Hasegawa was rescued by a fishing boat about 4 h later [14]. It should be noted here that the first tsunami wave did not influence or damage the first four sections of the Aonae urban area because they were protected by a hill separating the tsunami from the other part of the village. However, the tsunami branched and reflected as it moved around Aonae Bay (see Figure 1b).

22:22—A tsunami warning was issued by Japan Meteorological Agency (JMA).

22:23–22:25—Two fire engines were dispatched to provide tsunami warnings and evacuation advisories. At that time, the firefighters saw that the road near the harbor was flooded with seawater and that a car was floating.

22:24–22:26—After escaping from tsunami flooding up to her chest level by climbing onto the roof of her home in Hamatsumae (see Figure 1b), N. Iida witnessed what looked like steam or smoky mist rising from the seawater. T. Kikuchi, who witnessed the event from a hill above Aonae village at Point B where he had evacuated following the quake, said, "The offshore area looked pale white."

22:27—The NHK TV crew evacuated from the inn located around Point C in Figure 2a and started recording the boat fires around Point C approximately 10 min after the mainshock. In the video, the two closest fires appeared at two different locations inside and outside the seawall (boats #2, #3 and #4, #5 in Figure 2b and a, respectively). Then, after about 30 s from the screenshot shown in Figure 2b, flames from the four boats seemed to merge into two bigger fires (see Figure 1c). The video then recorded a boat moored near the nearest seawall catching fire, as shown in Figure 2d,e, at which point the fires grew violently. According to the NHK TV video narration, "One could not see the tsunami coming at that time." Note that although the first tsunami wave had branched and reflected within Aonae Bay, its destructive power had decreased and the only damage was from the rising sea level, as mentioned

above. Nevertheless, T. Kikuchi, observing from Point B in Figure 2a, also testified that "five moored boats burned" [14], which was consistent with the NHK TV video.

22:33—When gusts accompanying the second tsunami wave struck, the burning fishing boats were blown ashore and fires spread into the urban area (Figure 3a,b). T. Kikuchi testified as follows: "A strong wind arose and the burning boats were swept to the port interior. The fires ignited the gasoline of a damaged car that had been overturned by the second tsunami wave and eventually spread to the center of the urban area" [14] (see Figure 3a–c). S. Sakamoto, who also watched the events from Point D in Figure 2a from the moment the fire broke out said, "I heard eerie loud Goooo … Goooo … noises." When he looked toward the port where the noises seemed to be originating, he saw a hazy whitish wave that looked like it was burning above the water [14] (see Figure 3b). The inhabitants that evacuated to the hills surrounding the village reported witnessing a series of other strange tsunami-related fires.

22:40—Two more fire engines were dispatched for fire extinguishing activity from a fire station on the hill, but they were blocked by rubble from the tsunami and forced to return.

**July 13**

09:20—Fires in the urban area were extinguished. In the second to fourth districts of the Aonae urban area, a total 5.1 ha and 189 houses had been destroyed.

### 3.2. Working Hypothesis: Modeling the Tsunami Fires

We will now focus on investigating the possible mechanism that first triggered the moored boat fires and later the Aonae urban fires. Taking the above timeline into consideration, we can see that five boat fires occurred almost simultaneously, even though they were moored separately from each other (see Figure 2a–e). These fires were so intense that within just a few tens of seconds the four flames in Figure 2b merged into the two big fires shown in Figure 2c. The primary question then arises as to what could have caused the ignition of fire on five fishing boats that were positioned separately from each other so soon after the sea level rise caused by the first tsunami wave.

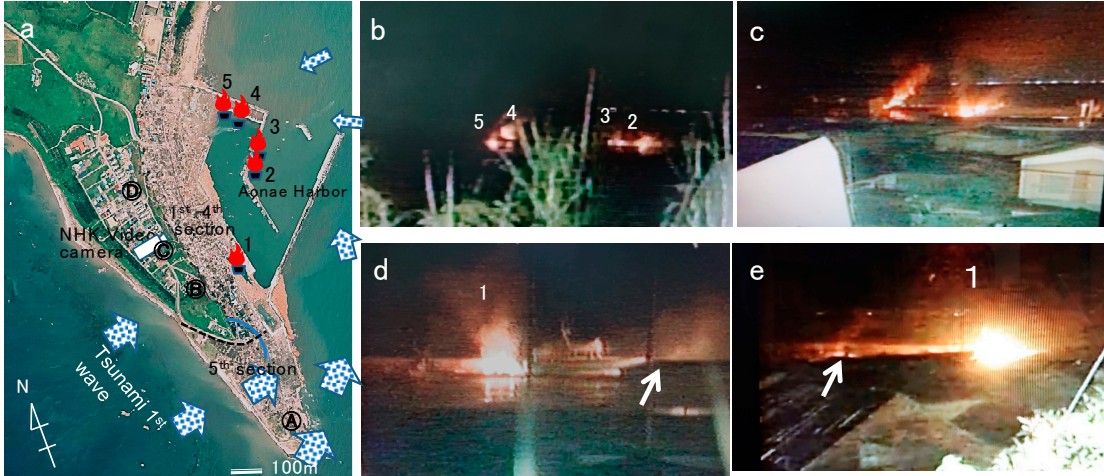

**Figure 2.** (**a**) Aerial photograph of Aonae district showing the locations of the boat fires that occurred when the first tsunami wave arrived (about 10 min after the mainshock). The photo was taken by the Geospatial Information Authority of Japan (GSI) on July 13, one day after the earthquake. Points Ⓐ–Ⓓ indicate locations where the boat fires were witnessed or recorded by the NHK (Japan Broadcasting Corporation). The numbers 1–5 indicate the boat fires. (**b**,**c**) Boat fires 2–5. (**d**,**e**) Two views of boat fire; arrows indicate whitish splashed waves and fiery hot spots, respectively. The source: The "NHK Special" program entitled "*Oo-tsunami ga osotta, Okushiri-tō karano Houkoku*" (in Japanese), broadcast on July 16, 1993.

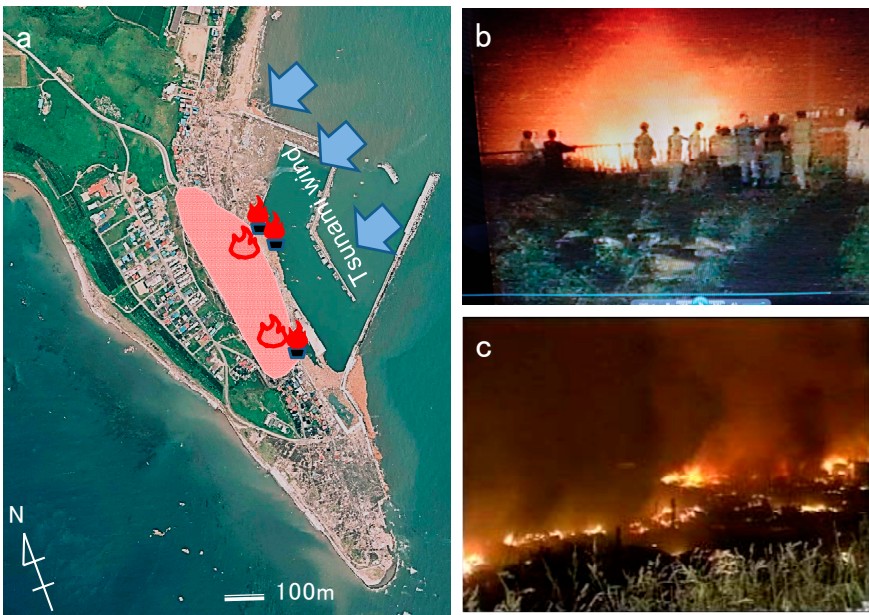

**Figure 3.** (**a**) Aerial photograph of the Aonae urban area (GSI) where Sections 1–4 were destroyed. (**b**,**c**) Views of urban area fires spread by gusts accompanying the second tsunami wave after the burning fishing boats were blown ashore. The source: The "NHK Special" program entitled "*Oo-tsunami ga osotta, Okushiri-tō karano Houkoku*" (in Japanese), broadcast on July 16, 1993.

In laboratory experiments simulating vehicle fires caused by a tsunami, it took from 25 min to 4 h for a battery-initiated fire to ignite after immersion in seawater [15]. Additionally, it must be noted that the diesel fuel used in boat engines is far less likely to ignite than gasoline used in cars, which makes it very unlikely that the boat fires occurred due to the seawater immersion of their marine diesel engine systems. Note that the fishing boats were not the only fires that could be seen around the seawall. In fact, fiery hot spots and shining splashes similar to those seen in the boat fires are indicated by arrows in Figure 2d,e. This implies that, even in the absence of inflammable materials such as those used in the urban area (gasoline, diesel, propane, etc.) some other flammable substance was burning.

Then, what was the other burning substance? The key to solving the mystery may lie in the whitish-foaming/bubbling seawater described in the abovementioned eyewitness testimony by Iida at her Hamatsumae residence. We posit that the whitish foam observed might be bubbles containing gaseous methane released from the seabed, as that substance, in the solid form of methane hydrate created by both bacterial and thermogenic processes acting on complex organic matter, might be abundantly present in the seabed off Okushiri Island.

### 3.3. Where Did Methane Release?

A survey conducted from 31 July to 7 September 1993 by the Japan Agency for Marine-Earth Science and Technology (JAMSTEC) found white bacterial mats on the slope fissure zone near the southern end of the tsunami generation area (a dotted square in Figure 1b), which indicate the presence of methane/methane hydrate [16]. Most of the bacterial mats were found to exist along a fissure/cracked zone running parallel to the slope strikes at depths in the range of 400–2200 m [16]. The velocity of the bubbles rising from the seabed depends upon the release depth, bubble size, and bubble surface substance (clean, dirty, etc.), but if considering the maximum as 30 cm/s for a clean bubble with the size of 500 microns [17], it would clearly take a significant amount of time for methane bubbles generated at depths of ~1000 m to rise to the sea level.

However, with the assistance of strong upwell seawater motions along with steep slope, methane bubbles generated in the upper area of the slope, say 400 m deep, might have risen fast enough to be captured by the first tsunami wave. Additionally, it should be noted that inorganic methane is

generated even in shallow sea sediments on the continental margins, especially in rapidly deposited muddy sediments with high organic matter content. Indeed, such gassy sediments are often found in river deltas, estuaries, and harbors [18,19].

### 3.4. How Were Methane Bubbles Carried to the Coastal Area?

Next, we will discuss how the methane bubbles could have been transported to the coast by the tsunami. According to Goddin [20], the enhancement of tsunami-induced wind velocity perturbations, in comparison with currents in the tsunami wave, occurs because of the generation of viscous waves in the air caused by a coherent elevation of large expanses of the ocean surface by the tsunami wave. Additionally, at 0.555, the specific gravity of methane (and even that of a methane bubble) is lighter than that of air (specific gravity = 1.000). Furthermore, the bubble film generated at the seabed would be contaminated with organic molecules, which would elongate the life of those bubbles.

The white-bubbled tsunami-accompanying mists at Aonae Bay, as witnessed by the Hamatsumae resident, were not recorded in the NHK TV video, but very similar scenes were found in NHK TV video footage of Sendai Bay (see Figure 1a) during the tsunami following the 2011 Tohoku-Oki earthquake, as shown in Figure 4a–c, which showed a whitish-bubbled sea surface suddenly appearing several kilometers away from the coast. Those bubbles blazed violently, as can be seen in Figure 4b, after which the tsunami waters with rising mists or fumes bearing the bubbles propagated toward the coastline, possibly pushed by the "wind enhancement effect" of tsunami-induced perturbations (see Figure 4c).

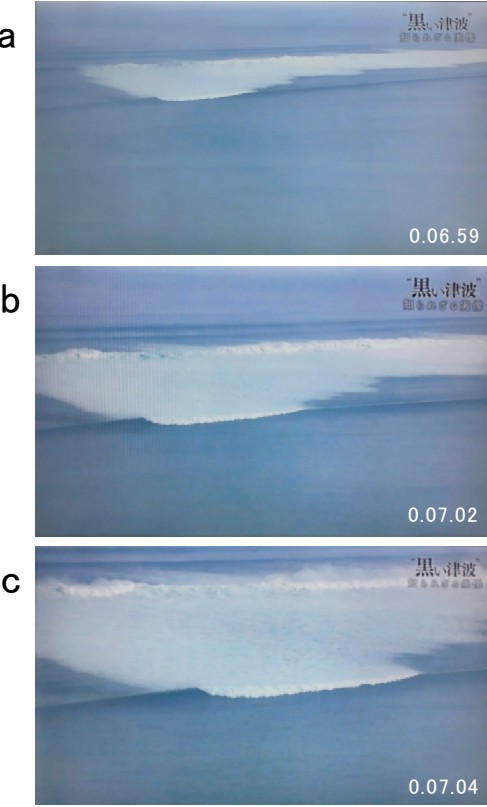

**Figure 4.** (**a–c**) Generation and propagation of whitish-bubbled tsunami water at 16:12 (local time) on 11 March as photographed in the 2011 Tohoku-Oki earthquake offshore of Sendai Bay, Natori city (see Figure 1a). Note that tsunami waters covered by smoke/mists are pushing numerous bubbles as they rush toward the shore. NHK TV video imagery from "Kuroi tsunami shirarezaru jituzo (in Japanese)" (Black tsunami, unknown images), which aired on March 3, 2019.

### 3.5. How Did the Methane Bubbles Ignite?

Next, we will consider what might have happened when the white-bubbled tsunami-accompanying mists hit the Aonae Harbor seawall. As shown in Figure 1b, a branched section of the first tsunami wave was reflected inside Aonae Bay. Although the tsunami was losing momentum, it struck the seawall forcefully and splashing can be seen in Figure 2d. Surface disruptions by processes such as splashing, bubbling, etc., can lead to charge separation and electrification [21]. While Bhattacharyya et al. showed that film droplets formed from bursting bubbles on the surface of pure water are predominantly negatively charged, primarily because the surface of pure water in their model had a slight excess of hydroxide ions [22], Blanchard has shown that the braking of air bubbles at an air–seawater interface can result in the production of positively electrified droplets [23].

In order to clarify some of the uncertainty related to film droplet electrification, we conducted laboratory experiments (Appendix A) in which a weight was dropped on seawater saturated with gaseous methane. The results showed that splashed mists/droplets were positively charged and that the electrification of seawater is more intense than that of deionized water, as shown in Figure 5a,b.

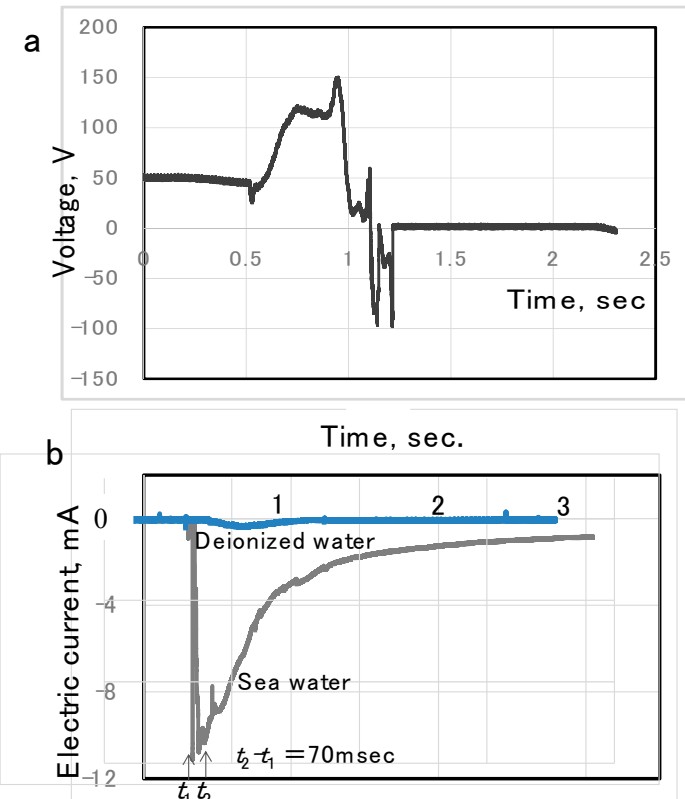

**Figure 5.** Results of the laboratory experiments of electrical response of splashed droplets when a steel weight was dropped on seawater bubbled with methane: (**a**) Electric potential of seawater, and (**b**) electric currents of de-ionized water (blue) and seawater (gray). Note that positive current is displayed on the negative side.

When positively charged mists are assumed to be suspended in the air at a maximum state for a time period of $\Delta t = t_2 - t_1 = 70$ ms, as shown in Figure 5b (see arrows), we can estimate the total charge.

$$Q = \sum\nolimits_{\Delta t} I(t) \delta t$$

where $I(t)$ is the current at the sampling time of $\delta t = 10$ μs during the time period $\Delta t$ of 70 ms as $Q = 6.45 \times 10^{-7}$ As. Taking the potential difference $V = 100$ V between the charged mists and the bottom seawater in Figure 5a, we can estimate the electrostatic energy

$$W = 1/2\, QV = 0.5 \times 6.4 \times 10^{-7} \text{ As} \times 100 \text{ V} = 0.032 \text{ mJ}$$

which is about one-tenth of the lowest minimum ignition energy (LMIE) of 0.30 mJ for methane [20]. In those experiments, therefore, methane could not be ignited.

However, the scale of the site, where a bright splash is observed in Figure 2d, as shown by the arrow, is more than 10 times larger than that of the present laboratory test system. Therefore, in large-scale fields (see Figure 2d,e), a large electrostatic energy, sufficient to ignite the gaseous methane from disrupted bubbles, might be generated between the sea surface and the water splashes ≥10 m high, as estimated from Figure 2d. The reason why the boats burned violently could be presumed to be that the methane-bearing foams flowed over the 3.5-m-high seawall and accumulated in large quantities on the boat decks.

Finally, it should be noted that electrostatic fires are disasters that sometimes occur in industrial processes that involve sprays of flammable liquids, such as painting, cooling, and washing (e.g., [24–27]). The present work suggested that similar phenomena could occur also in nature.

## 4. Concluding Remarks

From the results of the abovementioned discussion, we surmise that the physico-chemical processes leading to the tsunami-related fires that triggered the abovementioned fishing boat fires might be explained as resulting from the electrostatic ignition of gaseous methane released from disrupted bubbles. The total process is illustrated in Figure 6. It should also be noted that such tsunami-induced fires are not unique to the 1993 Hokkaido Nansei-Oki earthquake and the 2011 Tohoku-Oki earthquake. Looking through historical documents, we could find information that shows possible links to other tsunami-induced fires, such as those occurring during the 1896 Meiji-Sanriku tsunami, as shown in Appendix B, Figure A2. We consider it possible that the mechanisms proposed in this paper related to methane bubbles might be involved. It should also be noted that in the predicted Nankai Trough earthquake, tsunami-related fire risks are expected to exceed those of the 2011 Tohoku-Oki earthquake [9].

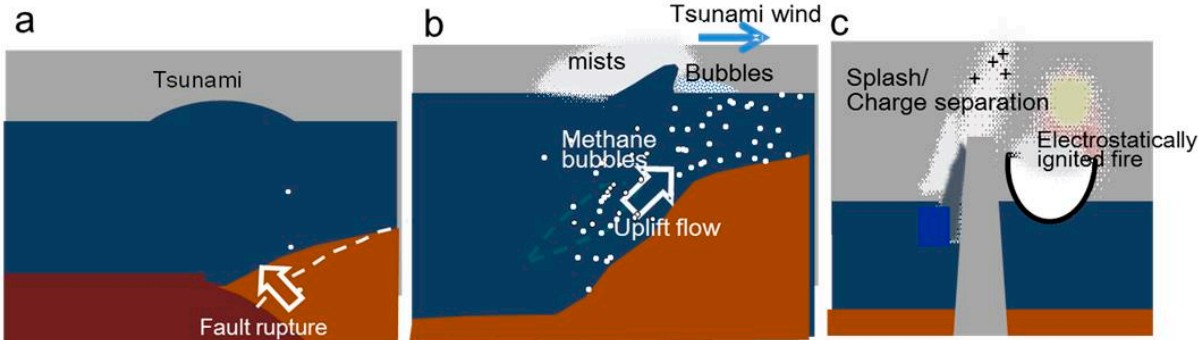

**Figure 6.** (**a**–**c**) Model showing how the boat fires could have started due to electrostatic ignition of methane-bearing bubbles (generated from the seabed due to the earthquake) striking the seawall.

**Author Contributions:** Conceptualization and methodology, Y.E.; experimental data curation, T.Y. and Y.E.; funding acquisition and project administration, S.S. and H.K.; writing—original draft preparation, Y.E.

**Acknowledgments:** We would like to thank the Okushiri Fire Station, Okushiri Town Hall, NHK Material Provision Business Department and Enterprise, Tohoku University Disaster Control Center, and the Geospatial Information Authority of Japan (GSI) for providing valuable materials and photographs related to the 1993 Hokkaido Nansei-Oki and 2011 Tohoku-Oki earthquakes. We would also like to extend our gratitude to Y.

Matsubara, E. Bernard, and two anonymous reviewers for their useful discussion and/or comments and to Genesis Research Institute, Inc. for providing research funds for this study.

**Conflicts of Interest:** The authors declare no conflicts of interest.

## Appendix A

We conducted laboratory experiments to verify the electrical phenomena associated with splashing seawater. Figure A1 shows a schematic drawing and photograph of our experimental setup. A 1.5 kg hollow steel cylinder, 110 mm in height and with outer and inner diameters of 45 and 22 mm, respectively, was dropped into seawater or deionized water suffused with gaseous methane from a height of 150 cm. A platinum-coated copper mesh electrode without electrical bias (i.e., floating potential) was set just above the height of the steel weight. During the splashing experiments, the resulting electrical potential or electrical current outputs were measured using a contact-type electrometer or a micro-ammeter.

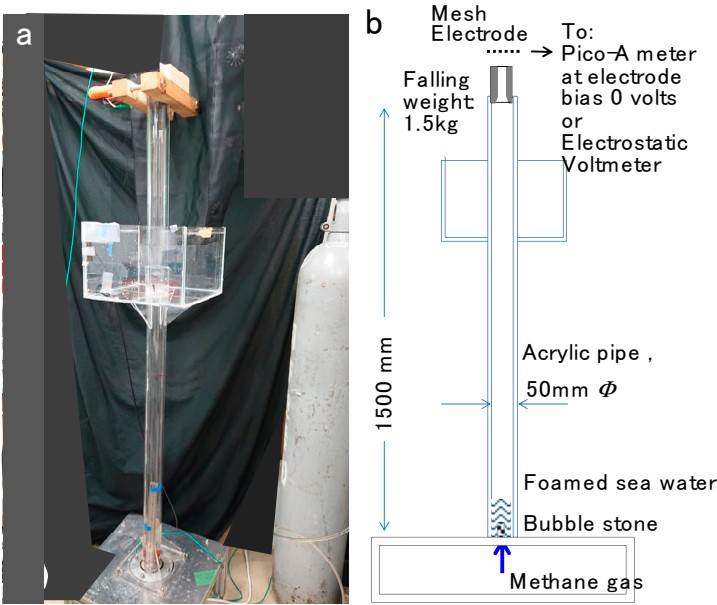

**Figure A1.** (**a**) Photograph and (**b**) schematic of experimental setup.

## Appendix B

The records of the 1896 M8.2–8.5 Meiji-Sanriku earthquake include a number of tsunami-related fires of unknown causes. According to the report "Iwate-ken Kaisho-shi" (in Japanese, unknown year of issue) one police officer witnessed dozens of lantern-size fires from houses up to the side of a nearby mountain at Ube village (see Figure 1b). Figure A2 shows a wooden print entitled the "Meiji-Sanriku Daikaisho no Jikkyo" (in Japanese, 1896) by M. Kokuni that shows a scene of a tsunami-stricken area. The left shows a submarine volcano eruption drawn based on the field explosion hypothesis. The right shows a flaming shrine that appears to be caused by a tsunami-related fire.

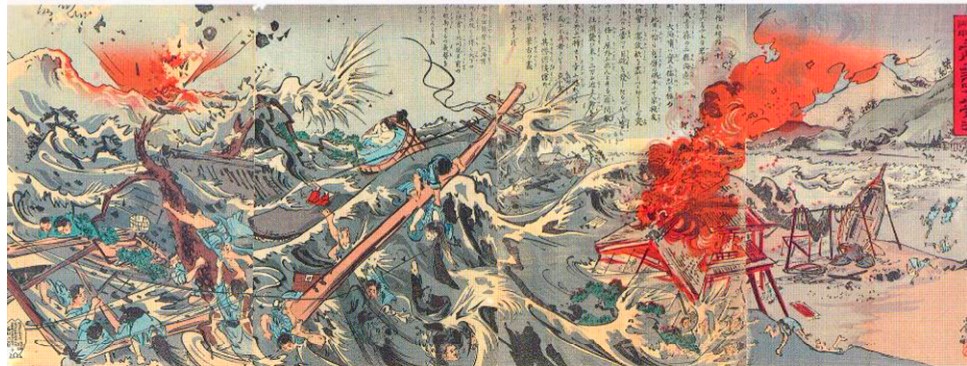

**Figure A2.** Wooden print (*Nishikie*) of the 1896 Meiji-Sanriku tsunami by M. Kokuni. (Holding of Y. Enomoto).

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
