# Peer review of "Possible Mechanism for the Tsunami-Related Fires That Occurred at Aonae Harbor on Okushiri Island in the 1993 Hokkaido Nansei-Oki Earthquake"

_geosciences, doi:10.3390/geosciences9060253_

Round 1
Reviewer 1 Report
This manuscript was very well written with an interesting hypothesis and work to back it up. A pleasure to read.
I have only two minor suggestions
line 50 rather than saying "after the tsunami" saying "after the fault rupture" is much clearer (twice in this line)
line 228 - the bracketed comment "(see fig 1a)" I would move this to line 226 directly after "Sendai Bay"
Author Response
Thank you very much for your valuable comments. Please check our reply in the attachment.

Reviewer 2 Report
This paper presents an interesting hypothesis for the occurrence mechanism of the tsunami-induced fires that occurred in the 1993 Hokkaido Nansei-Oki earthquake, Japan on the basis of video footage, testimonies, and experimental results. The paper is generally well-written and topic is clear presented. The main research findings of this paper will be important for full understanding of tsunami-induced fires and improving preparedness.
There is a concern that the introduction does not include references relevant to studies about tsunami-induced fires. As tsunami-induced fires occurred in great numbers in the 2011 Tohoku earthquake, the concern with tsunami-induced fires has been remarkably growing and several researchers have analyzed the occurrence tendency (e.g. Sekizawa and Sasaki, 2014), the damage situation and the hypothesis of occurrence mechanism (e.g. Hokugo et al., 2013), and have developed models for simulating the fires for improving preparedness (e.g. Nishino and Imazu, 2018). The fires presented in this paper seems to be different from various types of tsunami-induced fires that have been clarified by such studies. I recommended the authors overview relevant studies about tsunami-induced fires in the introduction section and refers to this paper’s uniqueness before I recommend for publication.
Sekizawa, A. and Sasaki, K., 2014. Study on Fires Following the 2011 Great East-Japan Earthquake based on the Questionnaire Survey to Fire Departments in Affected Areas, Fire Safety Science 11, 691-703.
Hokugo, A., Nishino, T. and Inada, T., 2013. Tsunami Fires After the Great East Japan Earthquake, Journal of Disaster Research 8, 584-593.
Nishino, T. and Imazu, Y., 2018. A Computational Model for Large-Scale Oil Spill Fires on Water in Tsunamis: Simulation of Oil Spill Fires at Kesennuma Bay in the 2011 Great East Japan Earthquake and Tsunami, Journal of Loss Prevention in the Process Industries 54, 37-48.
Author Response
Along with the reviewer 2's suggestion, "Introduction " are revised as noted in bule colored.
In addition, we add the description on the signal analyses of Fig.5 in 3.5 to make them more clear. Also noted in blue colored.
Revised manuscript is uploaded
Thank you all.

Round 2
Reviewer 2 Report
The manuscript has been improved on the basis of the review's suggestion. Therefore, this paper is recommended for publication. Minor revision is needed before publication because the survey report by the Japan Association for Fire Science and Engineering [13] is missing in the list of references.
Author Response
I add reference 13 as suggested by the reviewer.
Thank you for the comment.